# Investigation and Optimization of the Impact of Printing Orientation on Mechanical Properties of Resin Sample in the Low-Force Stereolithography Additive Manufacturing

**DOI:** 10.3390/ma15196743

**Published:** 2022-09-28

**Authors:** Enshuai Wang, Fei Yang, Xinmin Shen, Zhizhong Li, Xiaocui Yang, Xiangpo Zhang, Wenqiang Peng

**Affiliations:** 1College of Field Engineering, Army Engineering University of PLA, Nanjing 210007, China; 2State Key Laboratory of Disaster Prevention & Mitigation of Explosion & Impact, College of Defense Engineering, Army Engineering University, Nanjing 210007, China; 3Engineering Training Center, Nanjing Vocational University of Industry Technology, Nanjing 210023, China; 4College of Aerospace Science and Engineering, National University of Defense Technology, Changsha 410073, China

**Keywords:** additive manufacturing, low-force stereolithography, printing orientation, mechanical property, multiple regression model, parameter optimization

## Abstract

The mechanical properties of resin samples in low-force stereolithography additive manufacturing were affected by the printing orientation, and were investigated and optimized to achieve excellent single or comprehensive tensile strength, compressive strength, and flexural modulus. The resin samples were fabricated using a Form3 3D printer based on light curing technology according to the corresponding national standards, and they were detected using a universal testing machine to test their mechanical properties. The influence of the printing orientation was represented by the rotation angle of the resin samples relative to the *x*–axis, *y*–axis and *z*–axis, and the parameters was selected in the range 0°–90° with an interval of 30°. The multiple regression models for the mechanical properties of the prepared resin samples were obtained based on least square estimation, which offered a foundation from which to optimize the parameters of the printing orientation by cuckoo search algorithm. The optimal parameters for the tensile strength, compressive strength and flexural modulus were ‘*α* = 45°, *β* = 25°, *γ* = 90°’, ‘*β* = 0°, *β* = 51°, *γ* = 85°’ and ‘*α* = 26°, *β* = 0°, *γ* = 90°’, respectively, which obtained the improvements of 80.52%, 15.94%, and 48.85%, respectively, relative to the worst conditions. The mechanism was qualitatively discussed based on the force analysis. The achievements obtained in this study proved that optimization of the printing orientation could improve the mechanical properties of the fabricated sample, which provided a reference for all additive manufacturing methods.

## 1. Introduction

As a novel advanced manufacturing method, additive manufacturing has been broadly utilized in many industrial fields [1,2,3,4,5,6]. Additive manufacturing can deliver parts of very intricate and complex geometries with a need for minimum post-processing, built from tailored materials with near-zero material waste, while being applicable to a variety of materials, including plastics and metals [1,2]. In terms of materials processed, it has been reported that plastics were leading the additive manufacturing market, with around 30,000 machines in production [3]. The silica aerogel objects were fabricated by Zhao et al. [4], and the direct ink writing protocol was proposed to create the miniaturized silica aerogel objects from the slurry of silica aerogel powder in a dilute silica nanoparticle suspension. Additive manufacturing was also applied in the fabrication of the piezoelectric materials [5], which was favorable for developing the piezoelectric devices with an excellent performance, high energy storage density, and high electromechanical conversion efficiency. In addition, additive manufacturing technologies have shown the potential benefits of energy saving in the multiple energy sectors of nuclear energy, battery, fuel cell, oil and gas [6]. Therefore, many kinds of additive manufacturing methods have already been developed and some of them have been gradually introduced into commercial applications.

Among the present developed additive manufacturing methods, the stereo lithography apparatus (SLA) can be applied to form complex structured samples with high precision, and the utilization rate of the applied raw material is near 100% [7]. Moreover, SLA has the advantages of high efficiency and energy-saving [8], e.g., it can solidify the photosensitive resin completely within a few seconds at room temperature, with only 1/10–1/5 energy consumption compared with that of the conventional heat curing. Thus, SLA has been widely used in manufacturing dental prostheses by Yoo et al. [9], and the overall 3D accuracy of all 3D-printed resin models showed clinically acceptable ranges. A horn antenna with a 5th-order dual-mode ellipsoid resonator filtering network was developed by Liang et al. [10] through SLA, which achieved a lower loss and higher measured gain of antenna function. Zhao et al. [11] fabricated the periodical and ordered polymer–nickel-coated composite materials with a diamond-structure microlattice and various contents of phosphorus via electroless nickel–phosphorus (Ni–P) coating onto the diamond-structured polymeric templates using SLA, the maximum compressive strength of which was 2.1 times higher than that of polymer templates.

In order to further promote the actual application of the SLA, much research has been conducted to improve the mechanical properties of the fabricated resin sample [12,13,14,15,16,17,18,19,20]. Wang et al. [12] conducted topological optimization to achieve high-strength nodes in additive manufacturing by applying the bidirectional evolutionary structural optimization method. An AM-driven topology optimization method coupled with a transversely isotropic material model and a solid anisotropic material with penalization was proposed by Li et al. [13,14] to establish the quantitative correlation between the process-related parameters and mechanical properties of printed materials. Valencia et al. [15] studied the influences of the concentration of AgClO_4_, post-curing times, and other parameters on the performances of a metal/polymer nanocomposite with enhanced optical/electrical behavior. The topology optimization in the structural steel design for additive manufacturing was reviewed by Ribeiro et al. [16], which aimed to improve the performance of the fabricated sample. Martín-Montal et al. [17] studied the influence of certain printing parameters on printed material behavior, such as printing angles, printing resolution, curing time and temperature. Chen and Lu [18] proved that surface quality was not only dependent on build orientation, but more on scanning orientation of the parts in the rapid prototyping processes. It has been proven by Cazon et al. [19] that the part orientation has a significant effect on the elastic modulus and fracture stress in the PolyJet rapid prototyping technology. Udroiu [20] investigated the effects of the main factors in the material jetting additive manufacturing process on the surface roughness of the wing using the statistical design of experiments. These research achievements proved that adjustment of the process parameters in SLA and other additive manufacturing methods were the critical factors to decide the mechanical and other properties of the fabricated samples.

For most practical applications, the resin sample fabricated by SLA must have a certain mechanical property, which consists of the tensile strength, compressive strength, flexural modulus, and so on [21,22,23,24]. The printing parameters in SLA are critical factors to determine the mechanical properties of the resin samples, which can be divided into four kinds. First are the basic features of the photosensitive liquid resin, such as its composition, the proportion of each component, and the tailoring properties [25,26]. Second are the fabrication parameters in SLA, such as the wavelength and power of the laser, slice thickness, scanning interval, filling quantity, and environmental temperature and pressure [27,28]. Third are the parameters in the post-processing, such as the exposure time, type and power of the light source, and irradiation uniformity [29,30]. Fourth is the printing orientation, which consists of the rotation angles of the resin sample relative to the *x*–axis, *y*–axis and *z*–axis for the reference coordinate system in the SLA [18,19,20,31,32]. Furthermore, the printing parameters must be optimized to achieve the desired mechanical properties of the resin sample. Moreover, for the various desired mechanical properties, such as the tensile strength, compressive strength, or flexural modulus, the optimal parameters are different.

Relative to basic features of the photosensitive liquid resin [25,26], fabrication parameters [27,28], and parameters in the post-processing [29,30], studying the influence of the printing orientation is an easy but effective method to improve the mechanical properties of resin sample in SLA, which is propitious to promoting the SLA technology in the practical applications. Therefore, investigation and optimization of the impact of printing orientation on the mechanical properties of resin samples in SLA were conducted in this research. Taking the rotation angles of the resin sample relative to the *x*–axis, *y*–axis and *z*–axis for the reference coordinate system as the variables within the range of 0°–90° with the interval of 30°, the resin samples for the detections of tensile strength, compressive strength, and flexural modulus were fabricated by the SLA according to the corresponding national standards. Afterwards, the mechanical properties were detected by the universal testing machine, and the empirical model between the mechanical property and printing orientation was constructed to provide a foundation for the further optimization. Later, the parameters of the printing orientation were optimized for the single or comprehensive mechanical properties, and the effectiveness and accuracy of the optimization were further verified. Finally, influence of the printing orientation on the mechanism was investigated based on the printing principle in SLA and mechanical analysis, which also supplied references for other additive manufacturing methods. Above all, the aim of this study was to achieve optimal single or comprehensive mechanical properties through the preparation of samples with low-force stereolithography, the detection of mechanical properties with a universal testing machine, the construction of a theoretical model using the multiple regression method, and the optimization of parameters with the cuckoo search algorithm.

## 2. Materials and Methods

### 2.1. Material Preparation

The resin sample was fabricated using a Form3 low-force stereolithography 3D printer (Formlabs Inc., Boston, MA, USA) based on the light curing method according to the international standard of ‘ISO/ASTM52900–15 Standard Terminology for Additive Manufacturing–General Principles–Terminology’, as shown in Figure 1a. When fabrication of the sample was finished, it was further cleaned by the FormlabsForm wash (Formlabs Inc., Boston, MA, USA) to remove residual liquid resin and irradiated for solidification by the FormlabsForm cure (Formlabs Inc., Boston, MA, USA), as shown in Figure 1b,c, respectively. Except the printing orientation, all the other printing parameters were kept uniform. The photosensitive liquid resin used in this research was ClearV4, which was purchased from the self-support flagship store of Formlabs 3D printer via JD.com (JD.com Inc., Beijing, China). Furthermore, the fabrication parameters and the post-processing parameters were confirmed according to the operation manual of the utilized Form3 low-force stereolithography 3D printer based on the international standard of ‘ISO/ASTM52900–15 Standard Terminology for Additive Manufacturing–General Principles–Terminology’, summarized in Table 1 and Table 2, respectively.

A schematic diagram of the fabrication process in the SLA is shown in Figure 2, and it can be observed that the material forming process consisted of line-by-line scanning and layer-by-layer accumulation. For the initial statement, the photosensitive liquid resin was introduced into the resin tank, and the bottom of the lifting platform immersed into the liquid level. The thickness of each layer was controlled by the distance between the bottom of the lifting platform and the top surface of the transparent glass base plate. In the fabrication process, the laser spot scanned line-by-line to form a 2D section under the control of the computer program, and the photosensitive liquid resin in the exposure area was solidified rapidly. When one layer was finished, the lifting platform rose a distance for the thickness of the layer, and the fabricated resin sample was formed by this layer-by-layer accumulation.

Taking the fabrication capacity of the utilized Form3 3D printer into consideration, the sample for detection of tensile strength was selected and prepared at the reduction scale of 1:2 of the I-type sample according to the national standard of GB/T 1040–92 “Plastics–Determination of tensile properties”, and its actual photo is exhibited in Figure 3a. The middle sectional size of the width and thickness were 5 mm × 2 mm with the total length of 75 mm. The tensile strength *σ_b_* was chosen to characterize the tensile property of the resin sample in this study. Moreover, the sample for detection of the compressive strength was designed and prepared according to the national standard of GB/T 8813–2008 (ISO844: 2004, IDT) “Rigid cellular plastics–Determination of compression properties”, as shown in Figure 3b, and its length, width and thickness were 50 mm, 50 mm and 25 mm, respectively. The compression strength *σ_bc_* was selected to characterize the compression property of the resin sample in this research. Furthermore, the sample for detection of flexural modulus was designed and prepared according to the national standard of GB/T 9341–2008 (ISO178: 2001) “Plastics–Determination of flexural properties”, as shown in Figure 3c, and the length, width and thickness of the sample were 80 mm, 10 mm and 4 mm, respectively. The flexural modulus *E* was selected to characterize the flexural property of the sample.

The 3D model of the sample was constructed in the 3D modeling software Solidworks (Dassault Systèmes SOLIDWORKS Corp., Waltham, MA, USA) and it was saved as an .stl file. Afterwards, the .stl file was introduced into the Preform software supported by the Form3 3D printer, and its orientation could be adjusted relative to the reference coordinate system, as shown in Figure 4. The default directions for the 3 kinds of samples are shown in Figure 4a–c, respectively. The rotation angle of the resin sample relative to the *x*–axis, *y*–axis and *z*–axis was selected in the range of 0°–90° with the interval of 30°, which indicated that there were 64 group of samples (4 × 4 × 4) for each kind of detection of the mechanical properties. The circumstances of rotation 90° for the 3 kinds of samples are shown in Figure 4d–f, respectively, which correspond to rotation 90° relative to the *x*–axis, rotation 90° relative to the *y*–axis, and rotation 90° relative to the *z*–axis, successively. Furthermore, in order to reduce the accidental error and improve the detection accuracy, 5 samples were fabricated in this study for each group of parameters of the printing orientation, and the mechanical properties of each sample were detected separately. The final data on the mechanical properties were the arithmetical average values of 5 experimental results.

### 2.2. Sample Detection

The tensile strength and flexural modulus of the prepared resin samples were detected by the universal testing machine of WTB–5KN (Yangzhou Zhengyi Testing Machinery Co. Ltd., Yangzhou, China), and its schematic diagram is shown in Figure 5a. Taking the normal compressive strength of the resin sample into account, the compressive strength was detected by the universal testing machine of same brand with a larger testing force of 2000 KN. The detection process for the tensile strength, compressive strength, and flexural modulus was different, as shown in Figure 5b–d, respectively. 

The measurement procedures of the tensile strength for each resin sample were as follows. Firstly, the clamp for tensile testing was fixed and the positioning calibrated, which aimed to ensure the fitting accuracy between the upper part and lower part. Secondly, the universal testing machine was turned on and the position limits were installed, which aimed to ensure the safety of the laboratory staff and that of the equipment. Thirdly, the resin sample for tensile testing was fixed in the clamp, and the upper and lower edges of the electronic extensometer were bound to the sample with rubber bands. Fourthly, the matched software in the computer was opened and the plastic stretching procedure was selected. Moreover, the middle sectional size of the sample was input into the software, and the set system configuration was 1025E deformation sensor, automatic identification of fractures, preload force of 1 N, loading speed of 1 mm/min, and the full clearing of real-time data before the next time testing. Fifthly, when the sample was fractured or the load reached the maximum value, the testing process stopped. Finally, the test data were exported and the residual sample was taken down.

The measurement processes of the compressive strength for each resin sample were as follows. Firstly, the clamp for compressive strength testing was fixed and the positioning calibrated, which aimed to ensure the fitting accuracy between the upper part and lower part. Secondly, the universal testing machine was turned on and the position limits were fixed, which aimed to ensure safety of the laboratory staff and that of the equipment. Thirdly, the resin sample for compressive strength testing was laid flat on the center of the lower fixture, and the position of the upper fixture was adjusted close to the resin sample without any direct contact. Fourthly, the matched software in the computer was opened and the compressive procedure was selected. Moreover, the size of the sample was input into the software, and the set system configuration was 1025E deformation sensor, automatic identification of fractures, preload force of 1 N, loading speed of 2.5 mm/min, and the full clearing of real-time data before the next time testing. Fifthly, when the sample was crushed or the load reached the maximum value, the testing process stopped. Finally, the test data were exported and the residual sample was taken down.

The measurement procedures of the flexural modulus for each sample were as follows. Firstly, the clamp for flexural modulus testing was fixed and the positioning calibrated, which aimed to ensure the fitting accuracy between the upper part and lower part. Then, the span of the lower fixture was adjusted to 16 times the width of the sample according to the national standard. Secondly, the universal testing machine was turned on and the position limits were fixed, which aimed to ensure the safety of the laboratory staff and that of the equipment. Thirdly, the resin sample for flexural modulus testing was fixed on the lower fixture, and position of the upper fixture was adjusted close to the resin sample without any direct contact. Fourthly, the matched software in the computer was opened and the flexural procedure was selected. In addition, the size of the sample and the span were input into the software, and the set system configuration was 1025E deformation sensor, automatic identification of fractures, preload force of 1 N, loading speed of 2 mm/min, and the full clearing of real-time data before the next time testing. Fifthly, when the resin sample was broken or the load reached its maximum, the testing process stopped. Finally, the test data were exported and the residual sample was taken down.

In the detection process of tensile strength, the test procedure was strictly consistent with the GB/T 1040–92 “Plastics–Determination of tensile properties”, and the test velocity was set as 1 mm/min. In the detection process of compressive strength, the test procedure was strictly consistent with the GB/T 8813–2008 (ISO844: 2004, IDT) “Rigid cellular plastics–Determination of compression properties”, and the test velocity was set as 2.5 mm/min. In the detection process of flexural modulus, the test procedure was strictly consistent with the GB/T 9341–2008 (ISO178: 2001) “Plastics–Determination of flexural properties”, and the test velocity was set as 2 mm/min.

The rotation printing angles of the resin sample relative to the *x*–axis, *y*–axis and *z*–axis were labeled as *α*, *β* and *γ*, respectively. The prepared resin sample was installed on the corresponding platform in Figure 5 for the detection of mechanical properties, and the cross-sectional experimental results are shown in Figure 6a–c, which corresponded to the data for tensile strength with the group of parameters *α* = 30°, *β* = 0° and *γ* = 30°, the data for compressive strength with the group of parameters *α* = 0°, *β* = 30° and *γ* = 0°, and the data for flexural modulus with the group of parameters *α* = 90°, *β* = 60° and *γ* = 60°, respectively. Please note that these cross-sectional experimental results shown in Figure 5 are arithmetical average values of the detection result of the mechanical properties of the 5 prepared resin samples for each group of parameters of the printing orientation.

Based on the experimental data obtained in the detection process, the mechanical properties of the detected resin samples with different printing orientations were calculated through Equations (1) and (2) corresponding to tensile strength *σ_b_* and compressive strength *σ_bc_*, respectively. In Equation (1), *F_b_* is the maximum force during the yield stage and *S_o_* is the original middle sectional size 5 mm × 2 mm of the sample. For the cross-sectional experimental result of tensile strength in Figure 5a, it can be seen that the maximum force during the yield stage was 714.3 N, so the tensile strength *σ_b_* can be calculated as 71.43 MPa. In Equation (2), *P* is the maximum force during the compression process and *A* is the sectional size 50 mm × 50 mm of the sample in the compression direction. For the cross-sectional experimental result of compression strength in Figure 5b, it can be seen that the maximum force during the compression process was 3.156 × 10^5^ N, so the compressive strength *σ_bc_* can be derived as 126.24 MPa.
(1)$$\sigma_{b}=\frac{F_{b}}{S_{o}}$$
(2)$$\sigma_{bc}=\frac{P}{A}$$

For the flexural modulus *E*, this can be calculated by Equation (3). Here, *σ_fi_* (*i* = 1, 2) is the flexural stress corresponding to the defection *s_fi_* (*i* = 1, 2); *ε_f_*_1_ and *ε_f_*_2_ are 0.0005 and 0.0025, respectively, which were selected according to the GB/T 9341–2008 (ISO178: 2001) “Plastics–Determination of flexural properties”. The flexural stress *σ_f_*_1_ and *σ_f_*_2_ can be calculated by Equation (4). Here, *F* is the applied force; *L*, *b* and *h* are 80 mm, 10 mm and 4 mm, respectively, which correspond to the length, width and thickness of the sample in Figure 3c. The defection *s_fi_* (*i* = 1, 2) can be calculated by Equation (5). Thus, Formula (3) for the flexural modulus *E* can be converted to (6). The flexural modulus *E* can be calculated as 3675.46 MPa. The testing and analysis software in the universal testing machine can directly calculate the mechanical properties, which are summarized in Table 3. It should be noted that these data shown in Table 3 are arithmetical average values of 5 experimental results. The testing results proved that for these 5 resin samples for each group of parameters of the printing orientation, the obtained curves between load and deformation were basically consistent, and the derived differences among the 5 calculated values of each mechanical property were smaller than 4%.
(3)$$E=\frac{\sigma_{f2}-\sigma_{f1}}{\varepsilon_{f2}-\varepsilon_{f1}}$$
(4)$$\begin{array}{cc} \sigma_{fi}=\frac{3FL}{2bh^{2}} & \left(i=1,2\right) \end{array}$$
(5)$$\begin{array}{cc} s_{fi}=\frac{\varepsilon_{fi}L^{2}}{6h} & \left(i=1,2\right) \end{array}$$
(6)$$E=\frac{L^{3}}{4bh^{3}}\cdot \frac{F_{f2}-F_{f1}}{s_{f2}-s_{f1}}=\frac{L^{3}}{4bh^{3}}\cdot \frac{\Delta F}{\Delta s}$$

## 3. Modeling and Optimization

It can be seen from Table 3 that the mechanical properties of the prepared resin sample are obviously variant for the different parameters of printing orientations, no matter the tensile strength, compressive strength or flexural modulus. Taking the tensile strength, for example, the maximum value 76.60 MPa was achieved when the parameters of printing orientations were *α* = 60°, *β* = 30° and *γ* = 90°, and the minimum value 42.86 Mpa was obtained when the parameters of printing orientations were *α* = 90°, *β* = 90° and *γ* = 90°, which indicated that the deviation could reach 78.72% (78.72% = (76.60−42.86)/42.86 × 100%). In addition, the maximum value of the flexural modulus was 4520.86 MPa with the printing orientations of *α* = 30°, *β* = 0° and *γ* = 90°, and the minimum value was 3055.42 MPa with the printing orientations of *α* = 60°, *β* = 0° and *γ* = 0°, which indicated that the deviation reached 47.96% (47.96% = (4520.86−3055.42)/3055.42 × 100%). It can be judged from Table 3 that the mechanical properties are significantly affected by the printing orientation. Thus, in order to obtain the optimal mechanical properties, the theoretical model between the mechanical property and the printing orientation was essential to construct based on the experimental data. Afterwards, the parameters of printing orientation were optimized to achieve the best single or comprehensive mechanical properties.

### 3.1. Theoretical Modeling

In this study, there was no explicit functional relationship between the parameters of printing orientations and the mechanical properties of the prepared resin samples, and it was difficult to derive the accurate functional formula according to the printing mechanism in SLA. When there was no strict and definite functional relationship between the independent variable and dependent variable, the functional formula could be obtained quantitatively through the classical multiple regression equation [33,34], and its fundamental model is exhibited in Equation (7). Here, the regression coefficient *λ_i_* (*i* = 1, 2, …, *m*) and the deviation *ε* are the undetermined parameters, which have no relationship with the independent variable *x_i_* (*i* = 1, 2, …, *m*). Moreover, the deviation *ε* is in the standard normal distribution with the parameter of *σ*.
(7)$$\left\{ \begin{array}{l} y=\lambda_{0}+\lambda_{1}x_{1}+\cdots +\lambda_{m}x_{m}+\varepsilon \\ \varepsilon \sim N(0,\sigma^{2}) \end{array}\right.$$

The data for the sample set (*y_j_*, *x_j_*_1_, *x_j_*_2_, …, *x_jm_*) (*j* = 1, 2, …, *n*) were obtained by the experiments, and here *x_j_* is the variable and *y_j_* is the response value. Thus, the relationship in Equation (4) can be converted to Equation (8).
(8)$$\left\{ \begin{array}{l} Y=X\Gamma +Ε\\ \varepsilon_{i}\sim N(0,\sigma^{2}E_{n}) \end{array}\right.$$

Here $X=\left[\begin{array}{ccc} 1 & x_{11} & \cdots \\ \vdots & \vdots & \cdots \\ 1 & x_{n1} & \cdots \end{array}\begin{array}{c} x_{1m}\\ \vdots \\ x_{nm} \end{array}\right]$, $\Gamma ={\left[\begin{array}{cccc} \lambda_{1} & \lambda_{2} & \cdot \cdot \cdot & \lambda_{m} \end{array}\right]}^{T}$, $Y={\left[\begin{array}{cccc} \lambda_{1} & \lambda_{2} & \cdot \cdot \cdot & \lambda_{m} \end{array}\right]}^{T}$, $Ε={\left[\begin{array}{cccc} \varepsilon_{1} & \varepsilon_{2} & \cdot \cdot \cdot & \varepsilon_{m} \end{array}\right]}^{T}$, and *E_n_* was the unit matrix with *n* orders.

For the parameters *λ_i_* (*i* = 1, 2, …, *m*), they could be estimated using least square estimation, which aimed to minimize the sum of the quadratic error of prediction *Q*, as shown in Equation (9).
(9)$$Q=\sum_{i=1}^{n}\varepsilon_{i}^{2}=\sum_{i=1}^{n}{\left(y_{i}-\lambda_{0}-\lambda_{1}x_{i1}-\cdots -\lambda_{m}x_{im}\right)}^{2}$$

Besides the constant term *λ*_0_ and the primary term *λ_i_x_i_* (*i* = 1, 2, …, *m*), introduction of the high order terms and the cross terms could be conducive to improving the accuracy of the theoretical model. Therefore, the cubic higher order term and the cubic cross term were applied in the model of multiple regression equation, as shown in Equations (10)–(12), which correspond to the tensile strength *σ_b_*, compressive strength *σ_bc_* and the flexural modulus *E*, respectively.
(10)$$\sigma_{b}=\sum_{k=0}^{3}\sum_{j=0}^{3}\sum_{i=0}^{3}\lambda_{b-ijk}\alpha^{i}\beta^{j}\gamma^{k}$$
(11)$$\sigma_{bc}=\sum_{k=0}^{3}\sum_{j=0}^{3}\sum_{i=0}^{3}\lambda_{bc-ijk}\alpha^{i}\beta^{j}\gamma^{k}$$
(12)$$E=\sum_{k=0}^{3}\sum_{j=0}^{3}\sum_{i=0}^{3}\lambda_{E-ijk}\alpha^{i}\beta^{j}\gamma^{k}$$

The experimental data obtained with parameters of *α* = 0°, *β* = 30° and *γ* = 60° and those obtained with parameters of *α* = 90°, *β* = 0° and *γ* = 60° were selected as the testing samples, and the other experimental data were treated as the prediction set, which aimed to derive the theoretical model according to the multiple regression model. The two testing samples were randomly selected from the experimental data. Based on the least square estimation, the multiple regression models for the mechanical properties of the prepared resin samples were obtained, as shown in Equations (13)–(15), which correspond to tensile strength *σ_b_*, compressive strength *σ_bc_* and the flexural modulus *E*, respectively.
(13)$$\begin{array}{cc} \sigma_{b}= & 73.03-2.90\times 10^{-2}\alpha -5.10\times 10^{-2}\beta -1.99\times 10^{-1}\gamma \\ & +2.92\times 10^{-3}\alpha^{2}-1.61\times 10^{-3}\beta^{2}+1.23\times 10^{-3}\gamma^{2}+7.20\times 10^{-4}\alpha \beta +3.65\times 10^{-3}\alpha \gamma +4.90\times 10^{-3}\beta \gamma \\ & -2.90\times 10^{-5}\alpha^{3}-2.00\times 10^{-5}\beta^{3}+8.00\times 10^{-6}\gamma^{3}-1.60\times 10^{-5}\alpha^{2}\beta -3.10\times 10^{-5}\alpha^{2}\gamma \\ & -1.00\times 10^{-6}\alpha \beta^{2}-1.40\times 10^{-5}\alpha \gamma^{2}-1.50\times 10^{-5}\beta^{2}\gamma -4.50\times 10^{-5}\beta \gamma^{2} \end{array}$$
(14)$$\begin{array}{cc} \sigma_{bc}= & 121.81+1.42\times 10^{-1}\alpha +1.23\times 10^{-1}\beta +1.68\times 10^{-1}\gamma \\ & -1.10\times 10^{-3}\beta^{2}+2.94\times 10^{-3}\gamma^{2}-2.31\times 10^{-3}\alpha \beta -4.52\times 10^{-3}\alpha \gamma +5.90\times 10^{-4}\beta \gamma \\ & +9.00\times 10^{-6}\beta^{3}-3.40\times 10^{-5}\gamma^{3}+2.30\times 10^{-5}\alpha \beta^{2}+2.60\times 10^{-5}\alpha \gamma^{2}-2.50\times 10^{-5}\beta^{2}\gamma +1.20\times 10^{-5}\beta \gamma^{2} \end{array}$$
(15)$$\begin{array}{cc} E= & 3887-19.4\alpha +22.4\beta +10.4\gamma \\ & -1.18\times 10^{-1}\alpha^{2}-3.48\times 10^{-1}\beta^{2}-1.30\times 10^{-1}\gamma^{2}-8.90\times 10^{-2}\alpha \beta +1.75\times 10^{-1}\alpha \gamma -3.68\times 10^{-1}\beta \gamma \\ & +3.43\times 10^{-3}\alpha^{3}+9.40\times 10^{-4}\beta^{3}+7.00\times 10^{-4}\gamma^{3}-4.77\times 10^{-3}\alpha^{2}\gamma +1.44\times 10^{-3}\alpha \beta^{2}+3.00\times 10^{-3}\alpha \gamma^{2}\\ & +5.40\times 10^{-3}\beta^{2}\gamma -1.40\times 10^{-3}\beta \gamma^{2} \end{array}$$

The parameters of the two testing samples were introduced into the constructed multiple regression models and comparisons of prediction values with actual values were shown in Table 4. It can be observed that relative to the tensile strength *σ_b_* and flexural modulus *E*, the prediction accuracy of the compressive strength *σ_bc_* was higher, the relative errors of which were −0.69% and 4.84% for the two testing samples. The major reason for this phenomenon was that the number of the samples in the prediction set was not high enough, and the compressive strength *σ_bc_* was insensitive to the change of the parameters of the printing orientation relative to the tensile strength *σ_b_* or flexural modulus *E*.

Moreover, the obtained regression equations in Equations (13)–(15) were only for this resin and this low-force stereolithography method with the selected printing conditions, and these regression equations would be different if the resin or printing conditions were changed. However, the results of the experimental validation in Table 4 proved the effectiveness of the multiple regression model, which could provide meaningful guidance for the other printing conditions or the other additive manufacturing methods by adjusting the order and regression parameters in the multiple regression model.

### 3.2. Parameter Optimization

It could be judged from Table 3 that the mechanical properties of the prepared resin samples varied wildly for the different parameters of the printing orientation. Taking the tensile strength *σ_b_*, for example, the maximum value was 76.60 MPa with the parameters *α* = 60°, *β* = 30° and *γ* = 90°, and the minimum value was 42.86 MPa with the parameters *α* = 90°, *β* = 90° and *γ* = 90°. Therefore, it was essential to optimize the parameters of printing orientation to achieve the optimal mechanical properties. Moreover, it can be seen that the optimal parameters for various single mechanical properties were quite different, which were *α* = 60°, *β* = 30° and *γ* = 90° for the optimal tensile strength *σ_b_*, *α* = 0°, *β* = 60° and *γ* = 90° for the optimal compressive strength *σ_bc_*, and *α* = 30°, *β* = 0° and *γ* = 90° for the optimal flexural modulus *E*. Thus, for the actual applications which required excellent comprehensive mechanical properties, it was essential to optimize the parameters of printing orientation by taking into consideration the tensile strength *σ_b_*, compressive strength *σ_bc_*, and flexural modulus *E* simultaneously.

The cuckoo search algorithm was proposed and developed by Yang and Deb [35,36], and it performed a global search through simulating the parasitic brood behavior of the cuckoo nests using the Lévy flight, which has been widely utilized in various parameter optimization [37,38,39,40]. There are three major components in the Lévy flight, which consist of the random route, short–distance flight with a high frequency, and occasional long-distance flight, and its Pseudo codes are shown in Table 5 [37,38,39,40]. The short distance flight with a high frequency could lead to finding the optimal value in a small range during the solving process, and the occasional long-distance flight could avoid searching repeatedly near a local optimal solution and be propitious to obtaining a better value. Thus, the cuckoo search algorithm was selected to optimize the parameters of the printing orientation in this research, both in the optimization for the single mechanical property and that for the comprehensive mechanical properties.

#### 3.2.1. Optimization for Single Mechanical Property

Based on the obtained multiple regression models for the mechanical properties of the prepared resin samples in Equations (13)–(15), the parameters of printing orientation were optimized to achieve the best tensile strength *σ_b_*, compressive strength *σ_bc_*, and flexural modulus *E*, respectively. Taking the fabrication accuracy into consideration, the ranges of *α*, *β* and *γ* were selected as [0°, 90°] with the interval of 1°. Taking the optimization objective, empirical multiple regression models, and the constraint conditions in the calculation program of the cuckoo search algorithm, the optimal parameters of printing orientation for single mechanical properties could be obtained, which are exhibited in the following Section 4.

#### 3.2.2. Optimization for Comprehensive Mechanical Property

As mentioned above, the tensile strength *σ_b_*, compressive strength *σ_bc_*, and flexural modulus *E* could not achieve their optimal values simultaneously, which could also be judged from the experimental data of the mechanical properties for different printing orientations in Table 3. In order to meet the requirements of the comprehensive mechanical properties in some actual application conditions, the weighting method was utilized for the three investigated mechanical properties, as shown in Equation (16). Here, *S* is the comprehensive performance score; *k*_1_, *k*_2_, and *k*_3_ are the weight for the tensile strength *σ_b_*, the compressive strength *σ_bc_*, and flexural modulus *E*, respectively, and their sum is 1; *σ_b_*(*α*, *β*, *γ*), *σ_bc_*(*α*, *β*, *γ*), and *E*(*α*, *β*, *γ*) are the tensile strength, compressive strength and flexural modulus corresponding to the group parameters of the printing orientation (*α*, *β*, *γ*); max(*σ_b_*) and min(*σ_b_*) are the maximum and minimum values of the tensile strength *σ_b_* in Table 3; max(*σ_bc_*) and min(*σ_bc_*) are the maximum and minimum values of the compressive strength *σ_bc_* in Table 3; and max(*E*) and min(*E*) are the maximum and minimum values of the flexural modulus *E* in Table 3. The dimensionless treatment in Equation (16) aimed to eliminate influence of the absolute value for the different mechanical properties.
(16)$$S=k_{1}\frac{\sigma_{b}(\alpha ,\beta ,\gamma )-\min(\sigma_{b})}{\max(\sigma_{b})-\min(\sigma_{b})}+k_{2}\frac{\sigma_{bc}(\alpha ,\beta ,\gamma )-\min(\sigma_{bc})}{\max(\sigma_{bc})-\min(\sigma_{bc})}+k_{3}\frac{E(\alpha ,\beta ,\gamma )-\min(E)}{\max(E)-\min(E)}$$

## 4. Results and Discussion

### 4.1. Optimal Parameters

The optimal parameters for single mechanical properties were obtained, which are summarized in Table 6. Taking the minimum in the experiment in Table 3 as the reference, the improvements obtained by the parameter optimization were 80.52%, 15.94%, and 48.85% for tensile strength *σ_b_*, compressive strength *σ_bc_*, and flexural modulus *E*, respectively. It can be observed that the improvement was most significant for the tensile strength *σ_b_*, which indicated that it was more sensitive to the parameters of printing orientation. On the contrary, the improvement for the compressive strength *σ_bc_* was only 15.94%, and this could infer that it was insensitive to the parameters of printing orientation, which was consistent with the former analysis of prediction accuracy.

Furthermore, nine conditions were taken into consideration and treated as the optimization objective for comprehensive mechanical properties, and the distributions of weights are shown in Table 7. For the conditions 1, 2 and 3, the major optimization targets were the tensile strength *σ_b_*, compressive strength *σ_bc_*, and flexural modulus *E*, respectively. For the conditions 4, 5, and 6, two of the three mechanical properties were treated as the primary objectives and the other one was treated as a secondary objective. For the conditions 7, 8, and 9, the weights of three mechanical properties were close and one of them was slightly larger, which can be judged from Table 7. Based on the weighting model in Equation (16) and the multiple regression models in Equations (13)–(15), the optimal comprehensive performance score *S*, the optimal parameters of printing orientation and the corresponding comprehensive mechanical properties were obtained and are summarized in Table 6 as well.

The achievements obtained in this research are consistent with the similar conclusions drawn in past research studies [18,19,20,31,32]. For example, Saini et al. [31] investigated the effect of layer orientations on the different mechanical properties of an SLA-manufactured polymer material by testing specimens printed with different orientations, and five different orientations, i.e., 0°, 22.5°, 45°, 67.5° and 90° were utilized to fabricate the specimens for the analysis. The experimental results showed that the maximum tensile and compressive load were obtained by the specimens printed at an angle of 22.5° and 67.5°, respectively. The specimen printed at 67.5° orientation again had the highest flexural strength, whereas the specimen printed at 0° achieved the higher impact and fatigue strength. Furthermore, the finds obtained by Noid et al. [32] indicated that both printing orientation and aging affect the flexural strength of additive-manufactured specimens. Four printing orientations were used in that study [32], which were group occlusal (the occlusal surface pointing down towards the print platform), group vertical (the distal side of specimen was facing the print platform), group palatal (the palatal side of specimen was facing the print platform), and group diagonal (positioning at a 45° angle with the mesial side facing the print platform). On the basis of these findings in the literature [18,19,20,31,32], the parameters of printing orientation were investigated in the 3D reference coordinate system, and they were further optimized for single or comprehensive mechanical properties, which could achieve better mechanical properties for the fabricated resin samples and promote the practical applications of the additive manufacturing method of low-force stereolithography.

### 4.2. Experimental Validation

The resin samples were prepared according to the obtained optimal parameters in Table 6 and Table 7 by the experimental apparatus shown in Figure 1 according to the relevant national standards, and their mechanical properties were further detected by the experimental apparatus shown in Figure 5. For each group of parameters of the printing orientation, five resin samples were prepared, respectively, and their mechanical properties were detected separately, and the final data of each mechanical property were arithmetical average values of five experimental results. Comparisons of the optimal single mechanical properties in theory with those in actuality are shown in Table 8 and Table 9, which corresponded to the optimization results for the single mechanical properties and those of the optimization results for the comprehensive mechanical properties, respectively. It can be seen that the consistency between the theoretical data and experimental data was satisfactory, which certified the effectiveness of the utilized optimization algorithm and the accuracy of the constructed empirical model. Moreover, it can be observed that the error for the optimal comprehensive mechanical property was larger relative to that for the optimal single mechanical properties, because the error for the empirical model of the comprehensive performance score *S* in Equation (16) was in the coupling superposition of the errors of empirical models for each single mechanical property in Equations (13)–(15). Moreover, it can be observed that the maximum value for each mechanical property in Table 9 did not reach their maximum value in Table 8, which indicated that the optimal comprehensive mechanical property was the compromise of the three single mechanical properties with the various weight distributions.

### 4.3. Mechanism Analysis

According to the fabrication process in the SLA [41,42,43,44], it can be seen that the fabricated resin sample consisted of large amounts of microparticles, which were controlled by the size of the laser spot and the thickness of each layer, and the force analysis for a single microparticle is shown in Figure 7. The red block, yellow blocks, green blocks, and purple blocks represent the analyzed microparticle, the nearby microparticles within the line show the scanning direction, the nearby microparticles in the neighboring lines show feed direction, and the nearby microparticles in the adjacent layers show accumulation direction, respectively. It can be observed that the force *F*_il_ existed within each line, force *F*_nl_ among the neighboring lines, and force *F*_al_ among the adjacent layers. The mechanical properties of the fabricated resin sample were the vector superposition of these forces. Taking into account the fabrication process of SLA, the order of three kinds of forces from large to small was *F*_il_, *F*_nl_ and *F*_al_, sequentially. Moreover, the reference coordinate system of Preform software supported by the Form3 3D printer treated *x*–axis, *y*–axis and *z*–axis as the feed direction, scanning direction, and accumulation direction, respectively.

According to the optimal parameters for single mechanical properties in Table 6, the corresponding printing orientations in the Preform software are shown in Figure 8, which correspond to the optimal printing orientation for best tensile strength *σ_b_*, that for best compressive strength *σ_bc_*, and that for best flexural modulus *E*, respectively. It can be intuitively judged from Figure 8 that the optimal printing orientations for each single mechanical property were quite different, and they were further qualitatively discussed one by one in the following part based on the force analysis.

#### 4.3.1. Tensile Strength

The optimal printing orientations for the best tensile strength were *α* = 45°, *β* = 25° and *γ* = 90°, as shown in Figure 8a. It can be observed that the optimal printing orientation was a complex spatial location instead of a simple location along one direction in the coordinate system. Compared with the feed direction, scanning direction, and accumulation direction in Figure 7, it can be seen that the optimal sample for best tensile strength was formed by sideling accumulation of the resin lines and the resin lines were mainly along the scanning direction (*y*–axis). As mentioned above, the order of three kinds of forces from large to small was *F*_il_, *F*_nl_ and *F*_al_, sequentially, in Figure 7, so the optimal printing orientations for best tensile strength could produce the largest resultant force of the three kinds of component forces. The force *F*_il_ within each line was mainly determined by the photosensitive liquid resin. The improvement of scanning speed and the decrease in scanning interval in the reasonable ranges could raise the force *F*_nl_ among the neighboring lines. In addition, the decrease in slice thickness in a reasonable range could raise the force *F*_al_ among the adjacent layers as well. Therefore, the optimal printing orientations for the best tensile strength was mainly among the scanning direction and had some inclinations in the feed direction and accumulation direction, and its exact value was determined by the utilized photosensitive liquid resin and the selected process parameters.

Moreover, it can be judged from Table 3 that the worst tensile strength was 42.86 MPa, which corresponded to the resin sample with the printing orientation *α* = 90°, *β* = 90° and *γ* = 90°, as shown in Figure 9a. It can be seen that the major tensile force acted on the force *F*_al_ among the adjacent layers, and the *F*_il_ within each line and force *F*_nl_ among the neighboring lines hardly worked in action. As mentioned above, the order of three kinds of forces from large to small was *F*_il_, *F*_nl_ and *F*_al_, sequentially, in Figure 7, and that was the major reason for the resin sample with the printing orientation *α* = 90°, *β* = 90° and *γ* = 90° achieving the worst tensile strength.

#### 4.3.2. Compressive Strength

The optimal printing orientations for the best compressive strength were *α* = 0°, *β* = 51° and *γ* = 85°, as shown in Figure 8b. It can be seen that the rotation angle *β* = 51° relative to the y–axis was close to 45° and that *γ* = 85° relative to the *z*–axis was close to 90°. The optimal sample for best compressive strength was formed by the oblique superimposition of the scanning lines. During the compression process, the main destruction was generated in the neighboring lines, because the binding force *F*_nl_ among the neighboring lines was smaller than *F*_il_ within each line. Moreover, the adjacent layers would transmit the loaded compressive pressure layer-by-layer. Thus, the optimal printing orientations for best compressive strength could produce the largest resultant force of the three kinds of component forces. Moreover, it can be judged from Table 3 that the worst compressive strength was 122.92 MPa, which corresponded to the resin sample with the printing orientation *α* = 0°, *β* = 0° and *γ* = 0°, as shown in Figure 9b. Compared with the optimal printing orientations *α* = 0°, *β* = 51° and *γ* = 85°, the resin sample for worst printing orientations was obtained by the plane superimposition of scanning lines to form each layer and the vertical accumulation of plane layers to form the whole solid, which indicated that the oblique interleaving would be favorable to achieve a better compressive strength.

#### 4.3.3. Flexural Modulus

The optimal printing orientations for the best flexural modulus were *α* = 26°, *β* = 0° and *γ* = 90°, as shown in Figure 8c. In the flexural process, the resin sample could be treated as a beam supported at both ends, and the beam was mainly formed along the scanning direction (*y*–axis). Furthermore, the accumulation of single scanning lines in the beam was not plane in the *z*–axis direction, which rotated 26° to make the force transmission for one layer to another sideling instead of vertical. Relative to the vertical force transmission with the printing orientation of *α* = 0°, *β* = 0° and *γ* = 90°, the flexural modulus was improved from 4450.11 MPa to 4584.83 MPa, which proved that the optimal printing orientations for best flexural modulus could produce the largest resultant force of the three kinds of component forces. Furthermore, it can be judged from Table 3 that the worst flexural modulus was 3055.42 MPa, which was for the resin sample with the printing orientation *α* = 60°, *β* = 0° and *γ* = 0°, as shown in Figure 9c. Relative to the beam formed mainly along the scanning direction (*y*–axis) for the optimal printing orientations, that for the worst printing orientations was in the combination direction of the feed direction (*x*–axis) and accumulation direction (*z*–axis), which could be considered as the minimum resultant force for the three kinds of component forces.

### 4.4. Optimal Comprehensive Mechanical Property

It can be judged from Table 7 that the optimal printing orientations for best comprehensive mechanical properties had an obvious relationship with those optimal printing orientations for single mechanical properties in Table 6. For the conditions 1, 2 and 3 in particular, which aimed to achieve the optimization targets with one major mechanical property and two secondary mechanical properties, the theoretical optimal parameters of printing orientations exhibited fine consistencies with those obtained for the single mechanical properties. In fact, the optimizations of the printing orientations for single mechanical properties in Table 6 were special cases for those of the comprehensive mechanical properties. The weight distributions were *k*_1_ = 1, *k*_2_ = 0, *k*_3_ = 0 for the parameter optimization of printing orientations with the target of best tensile strength, *k*_1_ = 0, *k*_2_ = 1, *k*_3_ = 0 for the parameter optimization of printing orientations with the target of best compressive strength, and *k*_1_ = 0, *k*_2_ = 0, *k*_3_ = 1 for the parameter optimization of printing orientations with the target of best flexural modulus.

Moreover, it could be interesting to note that except for condition 4, the optimal printing orientations for the other five conditions from 5 to 9 in Table 9 were all around *α* = 26°, *β* = 0° and *γ* = 90°, which were the optimal printing orientations for best flexural modulus *E*. The corresponding comprehensive mechanical properties of resin samples in theory with these optimal parameters for best single mechanical property are summarized in Table 10. It can be observed that for the printing orientations *α* = 26°, *β* = 0° and *γ* = 90°, not only could the flexural modulus achieve its maximum value, but also the tensile strength could obtain an excellent value 75.33 MPa, which was close to its maximum value 77.37 MPa. Furthermore, the maximum and minimum values of compressive strength were 142.51 MPa and 122.92 MPa, respectively, the variation range was smaller relative to the other two mechanical properties. Those were the major reasons for the similar optimal parameters for the conditions from 5 to 9 in Table 9.

## 5. Conclusions

Through sample preparation by low-force stereolithography, experimental detection by universal testing machine, theoretical modeling with multiple regression model and parameter optimization by cuckoo search algorithm, the parameters of printing orientation in the additive manufacturing process were investigated and optimized to achieve excellent single or comprehensive tensile strength, compressive strength, and flexural modulus. The major achievements obtained in this research are as follows.

(1)Based on low-force stereolithography apparatus and a universal testing machine, the influences of the parameters of printing orientations on the mechanical properties of resin samples were investigated in this research, which aimed to obtain better single or comprehensive tensile strength, compressive strength and flexural modulus.(2)The multiple regression models for the mechanical properties of resin samples were constructed according to the experimental data, as shown in Equations (13)–(15). Furthermore, the optimal parameters of printing orientations were obtained through the cuckoo search algorithm for the best single mechanical property and for comprehensive mechanical properties, respectively, as shown in Table 6 and Table 7. The optimal parameters for single tensile strength, compressive strength and flexural modulus were ‘*α* = 45°, *β* = 25°, *γ* = 90°’, ‘*α* = 0°, *β* = 51°, *γ* = 85°’ and ‘*α* = 26°, *β* = 0°, *γ* = 90°’, respectively, which obtained the improvements of 80.52%, 15.94%, and 48.85% relative to the worst conditions.(3)The experimental validation confirmed the effectiveness of the constructed multiple regression models and the optimization method. Mechanism analysis based on the force analysis for variable conditions qualitatively revealed the reasons for the differences in mechanical properties, which was also a meaningful reference for other 3D printing methods.

As mentioned in the Section 1, there were many influencing factors in the additive manufacturing method besides the investigated parameters of printing orientation in this research, such as the basic features of the photosensitive liquid resin, fabrication parameters, and post-processing parameters. Moreover, there were other parameters for the additive manufacturing methods other than low-force stereolithography. Therefore, more comprehensive considerations of the influencing parameters in various additive manufacturing methods should be taken into account in the further research, which could be favorable to promoting the applications of the additive manufacturing method.

## Figures and Tables

**Figure 1 materials-15-06743-f001:**
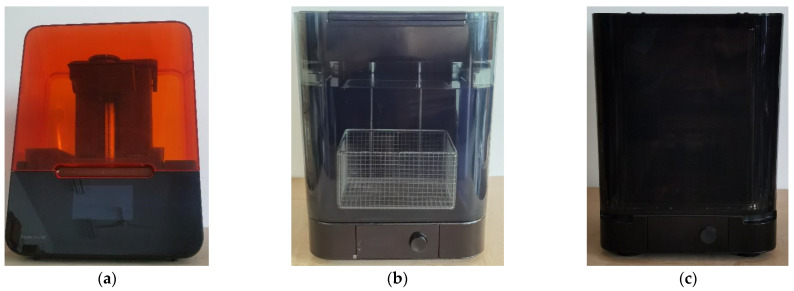
The apparatus for the material preparation. (**a**) Form3 low-force stereolithography 3D printer; (**b**) FormlabsForm wash; (**c**) FormlabsForm cure.

**Figure 2 materials-15-06743-f002:**
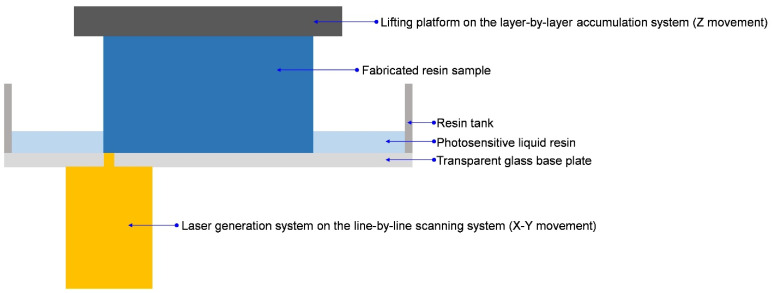
Schematic diagram of the fabrication process in the SLA.

**Figure 3 materials-15-06743-f003:**
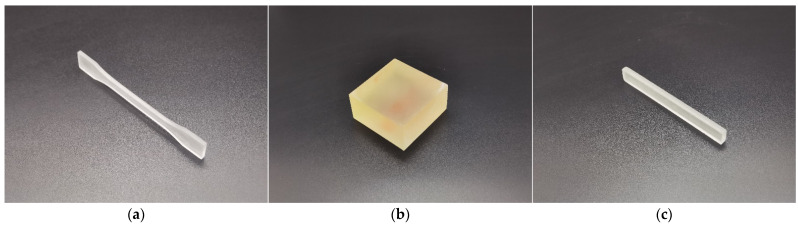
Actual photograph of the fabricated samples. (**a**) For the detection of tensile strength; (**b**) for the detection of compressive strength; (**c**) for the detection of flexural modulus.

**Figure 4 materials-15-06743-f004:**
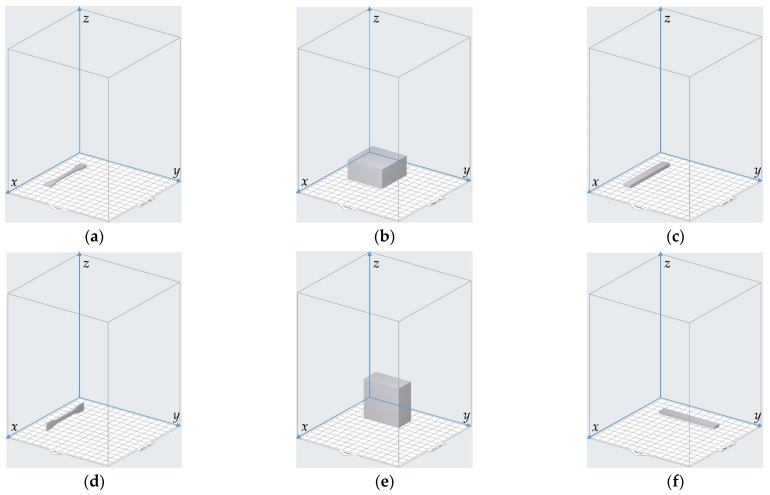
Adjustment of the printing orientations in the Preform software. (**a**) Default place of the sample for the detection of tensile strength; (**b**) default place of the sample for the detection of compressive strength; (**c**) default place of the sample for the detection of flexural modulus; (**d**) rotation 90° relative to the *x*–axis; (**e**) rotation 90° relative to the *y*–axis; (**f**) rotation 90° relative to the *z*–axis.

**Figure 5 materials-15-06743-f005:**
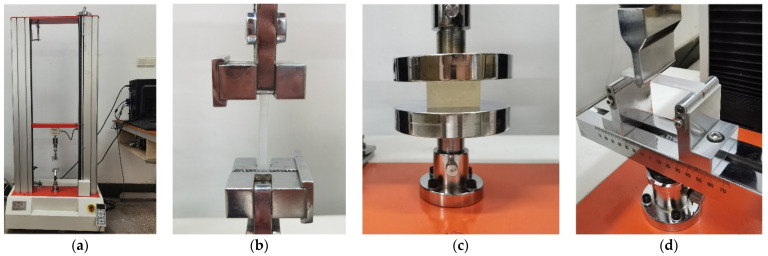
The detection of mechanical properties. (**a**) The universal testing machine; (**b**) detection of tensile strength; (**c**) detection of compressive strength; (**d**) detection of flexural modulus.

**Figure 6 materials-15-06743-f006:**
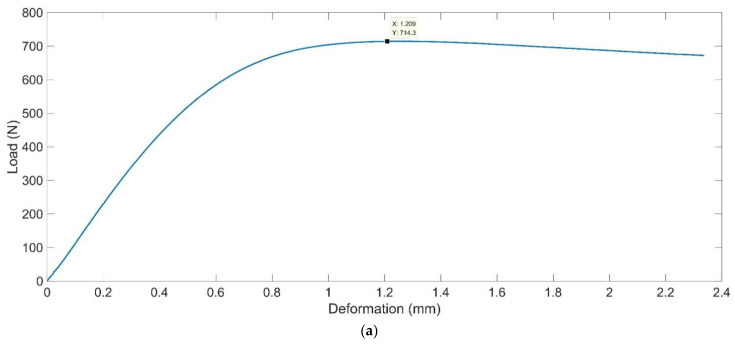
Cross-sectional experimental results of the mechanical properties of the prepared resin samples. (**a**) The tensile strength; (**b**) the compressive strength; (**c**) the flexural modulus.

**Figure 7 materials-15-06743-f007:**
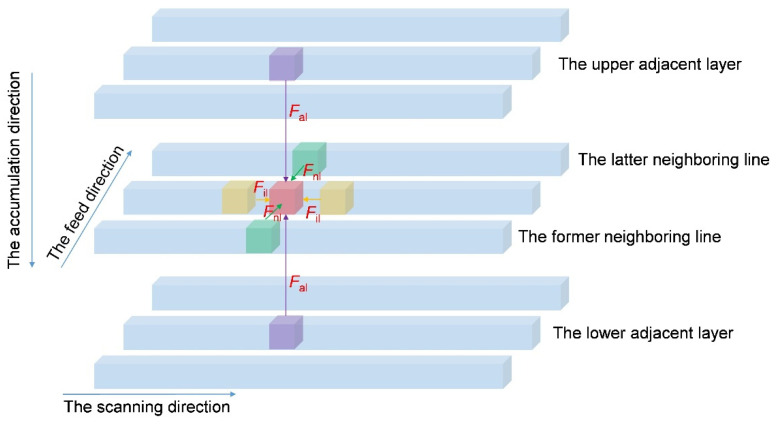
The force analysis for a single microparticle in the SLA.

**Figure 8 materials-15-06743-f008:**
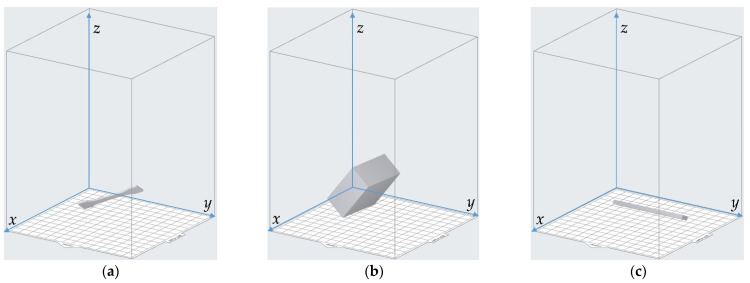
The optimal printing orientations for single mechanical properties in Preform software. (**a**) The optimal printing orientation for best tensile strength *σ_b_*; (**b**) the optimal printing orientation for best compressive strength *σ_bc_*; (**c**) the optimal printing orientation for best flexural modulus *E*.

**Figure 9 materials-15-06743-f009:**
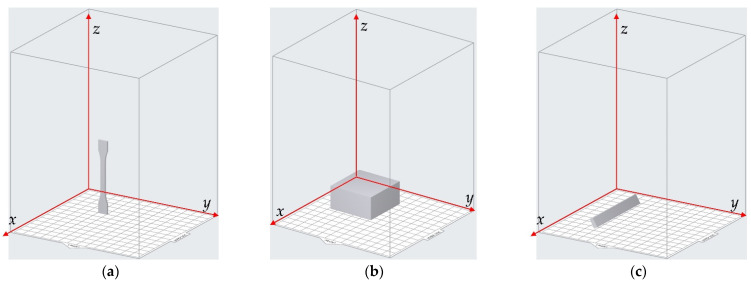
The printing orientations for the worst single mechanical property in Preform software. (**a**) The printing orientation for the worst tensile strength *σ_b_*; (**b**) the printing orientation for the worst compressive strength *σ_bc_*; (**c**) the printing orientation for the worst flexural modulus *E*.

**Table 1 materials-15-06743-t001:** The uniform fabrication parameters in this research.

Parameters	Values
Slice thickness	50 μm
Scanning interval	100 μm
Filling quantity	100%
Wavelength of the laser	C1-type laser product with wavelength of 405 nm
Power of the laser	250 mW power and laser facula with diameter of 85 μm
Operating temperature and its control	35 °C with heated air in the printing room
Environmental temperature and pressure	26 °C and 1.01 × 10^5^ Pa

**Table 2 materials-15-06743-t002:** The uniform parameters in the post-processing in this research.

Parameters		Values
In washing	Washing time	20 min
	Washing liquid	Isopropyl alcohol with concentration of 90%
In irradiation	Irradiation laser	Wavelength of 405 nm and power of 100 W
	Irradiation temperature, time and angle	60 °C, 30 min and 60°

**Table 3 materials-15-06743-t003:** The experimental data of mechanical properties for different printing orientations.

Serial Number	Parameters			Experimental Data		
	*α* (°)	*β* (°)	*γ* (°)	Tensile Strength (MPa)	Compressive Strength (MPa)	Flexural Modulus (MPa)
1	0	0	0	72.00 ± 1.21	122.92 ± 2.13	3746.87 ± 68.45
2	0	0	30	70.05 ± 1.33	128.62 ± 2.32	4205.82 ± 75.87
3	0	0	60	64.95 ± 1.18	136.14 ± 2.38	3904.87 ± 43.12
4	0	0	90	75.13 ± 1.44	133.61 ± 2.09	4450.11 ± 56.54
5	0	30	0	71.02 ± 1.05	126.24 ± 2.11	4458.46 ± 72.77
6	0	30	30	73.68 ± 1.19	131.59 ± 2.45	4236.11 ± 57.40
7	0	30	60	68.01 ± 0.96	140.04 ± 2.69	3943.97 ± 32.45
8	0	30	90	73.28 ± 1.27	138.07 ± 2.51	3770.86 ± 69.56
9	0	60	0	68.40 ± 0.87	127.37 ± 2.22	3935.85 ± 45.23
10	0	60	30	68.30 ± 0.91	131.38 ± 2.37	4357.97 ± 79.66
11	0	60	60	67.50 ± 0.82	140.97 ± 2.55	3898.84 ± 52.73
12	0	60	90	69.92 ± 1.07	142.21 ± 2.54	3771.12 ± 38.39
13	0	90	0	71.84 ± 1.16	129.24 ± 2.07	3728.53 ± 47.34
14	0	90	30	67.58 ± 0.88	131.30 ± 2.52	4487.99 ± 72.55
15	0	90	60	73.89 ± 1.46	139.22 ± 1.87	4193.04 ± 56.48
16	0	90	90	53.64 ± 0.73	141.54 ± 2.63	3975.76 ± 51.81
17	30	0	0	72.05 ± 1.34	126.01 ± 2.33	3435.50 ± 68.68
18	30	0	30	71.45 ± 1.29	135.22 ± 2.64	3446.22 ± 37.29
19	30	0	60	71.80 ± 1.12	137.21 ± 2.41	3907.05 ± 41.98
20	30	0	90	70.65 ± 1.23	133.95 ± 2.26	4520.86 ± 76.74
21	30	30	0	76.15 ± 1.46	132.99 ± 1.54	3480.88 ± 36.11
22	30	30	30	75.40 ± 1.41	136.45 ± 1.89	3668.20 ± 67.26
23	30	30	60	75.35 ± 1.29	139.21 ± 1.79	3678.11 ± 29.73
24	30	30	90	74.25 ± 1.23	131.55 ± 2.46	3902.00 ± 53.62
25	30	60	0	76.35 ± 1.38	126.66 ± 1.89	3447.95 ± 38.66
26	30	60	30	76.15 ± 1.51	136.02 ± 1.59	3724.32 ± 43.72
27	30	60	60	75.05 ± 1.44	134.56 ± 2.42	3840.17 ± 35.47
28	30	60	90	74.00 ± 1.43	138.57 ± 1.76	3966.77 ± 76.81
29	30	90	0	64.68 ± 1.22	127.51 ± 2.35	3401.35 ± 66.21
30	30	90	30	71.84 ± 1.41	133.83 ± 1.52	3728.53 ± 22.44
31	30	90	60	67.58 ± 1.07	138.23 ± 2.25	4487.99 ± 76.81
32	30	90	90	73.89 ± 1.17	135.64 ± 2.54	4193.04 ± 56.88
33	60	0	0	74.05 ± 1.29	133.66 ± 1.85	3055.42 ± 61.01
34	60	0	30	71.35 ± 1.20	127.55 ± 2.28	3308.51 ± 35.74
35	60	0	60	73.40 ± 1.35	127.67 ± 2.44	3569.08 ± 46.53
36	60	0	90	74.00 ± 1.41	124.34 ± 2.01	4323.24 ± 71.14
37	60	30	0	75.20 ± 1.48	132.71 ± 1.75	3490.59 ± 26.17
38	60	30	30	76.25 ± 1.50	128.25 ± 2.24	3248.06 ± 45.17
39	60	30	60	76.20 ± 1.22	137.05 ± 1.50	3241.02 ± 57.42
40	60	30	90	76.60 ± 1.38	128.76 ± 2.05	3609.35 ± 32.29
41	60	60	0	74.65 ± 1.29	130.66 ± 2.34	3453.00 ± 67.67
42	60	60	30	74.75 ± 1.19	135.09 ± 1.81	3218.02 ± 27.36
43	60	60	60	76.35 ± 1.42	130.17 ± 2.27	3378.50 ± 52.92
44	60	60	90	74.40 ± 1.31	125.35 ± 2.49	3672.22 ± 38.51
45	60	90	0	64.22 ± 0.91	134.57 ± 2.56	3276.81 ± 42.25
46	60	90	30	64.68 ± 0.88	127.00 ± 1.74	3692.52 ± 28.45
47	60	90	60	71.84 ± 1.08	125.37 ± 2.03	3728.53 ± 63.22
48	60	90	90	67.58 ± 0.91	133.63 ± 1.58	4487.99 ± 79.46
49	90	0	0	70.72 ± 1.37	132.38 ± 2.33	3742.75 ± 64.11
50	90	0	30	68.92 ± 1.10	132.34 ± 1.83	3334.49 ± 37.76
51	90	0	60	71.03 ± 1.32	129.85 ± 2.22	3929.58 ± 20.11
52	90	0	90	70.98 ± 1.14	134.23 ± 2.48	3964.02 ± 51.94
53	90	30	0	68.04 ± 1.36	133.61 ± 1.73	3894.56 ± 34.30
54	90	30	30	71.20 ± 1.29	130.14 ± 1.55	3488.27 ± 68.05
55	90	30	60	62.24 ± 0.97	130.88 ± 2.58	3365.84 ± 41.38
56	90	30	90	64.21 ± 0.74	132.43 ± 2.29	3429.32 ± 56.62
57	90	60	0	64.64 ± 0.47	134.10 ± 1.64	4016.64 ± 72.08
58	90	60	30	71.88 ± 1.25	128.80 ± 2.32	3391.80 ± 26.89
59	90	60	60	61.07 ± 0.68	132.30 ± 1.82	3675.46 ± 47.82
60	90	60	90	68.55 ± 1.12	133.41 ± 2.47	3591.54 ± 34.87
61	90	90	0	65.66 ± 0.82	141.54 ± 1.71	3975.76 ± 52.87
62	90	90	30	64.22 ± 0.74	139.22 ± 2.59	3771.12 ± 47.10
63	90	90	60	64.68 ± 0.81	131.30 ± 2.20	3770.86 ± 63.14
64	90	90	90	42.86 ± 0.53	129.24 ± 2.31	4450.11 ± 76.94

**Table 4 materials-15-06743-t004:** Comparisons of prediction values with actual values.

Parameters			Tensile Strength *σ_b_* (MPa)			Compressive Strength *σ_bc_* (MPa)			Flexural Modulus *E* (MPa)		
*α* (°)	*β* (°)	*γ* (°)	Actual	In Theory	Error	Actual	In Theory	Error	Actual	In Theory	Error
0	30	60	68.01 ± 0.96	69.78	2.60%	140.04 ± 2.69	139.08	−0.69%	3943.97 ± 32.45	4056.38	2.85%
90	0	60	71.03 ± 1.32	67.26	−5.31%	129.85 ± 2.22	136.14	4.84%	3929.58 ± 20.11	3591.65	−8.60%

**Table 5 materials-15-06743-t005:** Pseudo codes of the standard cuckoo search algorithm.

Cuckoo Search Algorithm
1: Objective function $f\left(x\right),x={\left(x_{1},x_{2},\ldots ,x_{d}\right)}^{T}$;
2: generate initial population of n host nests $x_{i}\left(i=1,2,\ldots ,n\right)$;
3: while (t≤MaxGeneration) or (stop criterion) do
4: obtain a cuckoo (say, i) randomly and generate a new solution by Levy flights;
5: evaluate its quality/fitness $F_{i}$;
6: choose a nest among n (say, j) randomly;
7: if ($F_{i}\geq F_{j}$) then
8: replace j with the new solution;
9: end if
10: a fraction ($P_{a}$) of worse nests are abandoned and new ones are built at new locations;
11: keep the best solutions (or nests with quality solutions);
12: rank the solutions and find the current best;
13: end while
14: post-process results and visualization.

**Table 6 materials-15-06743-t006:** The optimal parameters for single mechanical properties.

Optimization Objective	Optimal Parameters			Optimal Value	Minimum in Experiment	Improvement
	*α* (°)	*β* (°)	*γ* (°)			
Tensile strength *σ_b_* (MPa)	45	25	90	77.37	42.86 ± 0.53	80.52%
Compressive strength *σ_bc_* (MPa)	0	51	85	142.51	122.92 ± 2.13	15.94%
Flexural modulus *E* (MPa)	26	0	90	4548.08	3055.42 ± 61.01	48.85%

**Table 7 materials-15-06743-t007:** The optimal parameters for comprehensive mechanical properties.

Condition	Weight Distribution			Optimal *S*	Optimal Parameters			Comprehensive Mechanical Property		
	*k* _1_	*k* _2_	*k* _3_		*α* (°)	*β* (°)	*γ* (°)	*σ_b_* (MPa)	*σ_bc_* (MPa)	*E* (MPa)
1	0.6	0.2	0.2	0.8930	39	0	90	76.65	129.19	4456.67
2	0.2	0.6	0.2	0.8439	0	46	79	70.42	142.28	3791.37
3	0.2	0.2	0.6	0.9075	27	0	90	75.46	130.51	4506.38
4	0.4	0.4	0.2	0.8324	24	26	90	75.74	134.85	4012.15
5	0.4	0.2	0.4	0.8960	32	0	90	76.05	129.88	4493.05
6	0.2	0.4	0.4	0.8308	25	0	90	75.20	130.80	4508.05
7	0.4	0.3	0.3	0.8570	32	0	90	76.05	129.88	4493.05
8	0.3	0.4	0.3	0.8233	28	0	90	75.59	130.38	4504.71
9	0.3	0.3	0.4	0.8620	29	0	90	75.71	130.25	4502.52

**Table 8 materials-15-06743-t008:** Comparisons of optimal single mechanical property in theory with that in actuality.

Optimization Objective	Optimal Parameters			Theoretical Value (MPa)	Experimental Data (MPa)	Error
	*α* (°)	*β* (°)	*γ* (°)			
Tensile strength *σ_b_*	45	25	90	77.37	76.87 ± 1.17	0.65%
Compressive strength *σ_bc_*	0	51	85	142.51	144.32 ± 2.64	−1.25%
Flexural modulus *E*	26	0	90	4548.08	4584.83 ± 69.35	−0.016%

**Table 9 materials-15-06743-t009:** Comparisons of optimal comprehensive mechanical property in theory with that in actuality.

	Optimal Parameters			Comprehensive Mechanical Property								
				*σ_b_* (MPa)			*σ_bc_* (MPa)			*E* (MPa)		
	*α* (°)	*β* (°)	*γ* (°)	In Theory	Actual	Error	In Theory	Actual	Error	In Theory	Actual	Error
1	39	0	90	76.65	73.67 ± 1.47	4.05%	129.19	131.80 ± 2.58	−1.98%	4456.67	4498.01 ± 46.29	−0.92%
2	0	46	79	70.42	68.18 ± 1.33	3.29%	142.28	140.27 ± 1.83	1.43%	3791.37	3909.83 ± 57.72	−3.03%
3	27	0	90	75.46	71.14 ± 1.41	6.07%	130.51	133.87 ± 2.17	−2.51%	4506.38	4565.81 ± 72.33	−1.30%
4	24	26	90	75.74	74.17 ± 1.38	2.12%	134.85	130.99 ± 2.46	2.95%	4012.15	3895.12 ± 63.86	3.00%
5	32	0	90	76.05	71.71 ± 1.22	6.05%	129.88	133.14 ± 1.79	−2.45%	4493.05	4540.85 ± 59.41	−1.05%
6	25	0	90	75.20	72.51 ± 1.19	3.71%	130.80	134.03 ± 2.33	−2.41%	4508.05	4579.36 ± 77.57	−1.56%
7	32	0	90	76.05	71.71 ± 1.34	6.05%	129.88	133.14 ± 1.95	−2.45%	4493.05	4540.85 ± 81.15	−1.05%
8	28	0	90	75.59	70.96 ± 1.02	6.52%	130.38	133.23 ± 2.55	−2.14%	4504.71	4553.93 ± 54.98	−1.08%
9	29	0	90	75.71	70.69 ± 1.28	7.10%	130.25	133.78 ± 2.62	−2.64%	4502.52	4546.55 ± 67.64	−0.97%

**Table 10 materials-15-06743-t010:** The corresponding comprehensive mechanical properties of resin samples in theory with these optimal parameters for the best single mechanical property.

*α* (°)	*β* (°)	*γ* (°)	Tensile Strength *σ_b_* (MPa)	Compressive Strength *σ_bc_* (MPa)	Flexural Modulus *E* (MPa)
45	25	90	77.37	132.23	3903.64
0	51	85	70.19	142.51	3706.30
26	0	90	75.33	130.66	4548.08

## Data Availability

The data that support the findings of this study are available from the corresponding author upon reasonable request.

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
