# Peer review of "Investigation and Optimization of the Impact of Printing Orientation on Mechanical Properties of Resin Sample in the Low-Force Stereolithography Additive Manufacturing"

_materials, 2022, doi:10.3390/ma15196743_

Round 1

Reviewer 1 Report

In this paper, the authors investigate the mechanical properties of resin sample for different orientation, obtained by Low Force Stereolithography (LFS) on Form3 3D printer. Using the cuckoo search algorithm it was found the optimal part orientations with the best mechanical properties.

From my point of view there are some aspects to improve:

1.      The title of the paper should be written according to the 3d printing process that was investigates in the paper. Instead of Stereolitography should be used Low Force Stereolithography to avoid confusion with traditional Stereolitography process.

2.      Abstract does not contain any quantitative data about the results. The abstract should be revised and improved.

3.      The structure of the paper is unclear for a good understanding of the reader. Thus the paper structure should be revised in order to clearly present the Material and Methods, and Results in separated sections. The Material and Method section should describe clearly the materials used and the proposed methods. The section Results should shown all the results obtained by applying the proposed method. Thus, Figure 5, Table 3, 4, 5 and 6 present results. Figure 6 and 7 describe the LFS process and they should be moved to the Material and Method section.  

4.      The Introduction section should be improved. It should define the purpose of the work and its significance. The current state of the research field should be carefully reviewed and key publications cited. Please highlight controversial and diverging hypotheses when necessary. Which are the conclusions from the literature survey?

5.      Line 34: “The metal additive manufacturing was..”. Please explain if there are any link of the metal AM to this research.

6.      Line 80: “For most of practical applications, the resin sample fabricated by SLA must have a certain mechanical property, which consists of the tensile strength, compressive strength, flexural modulus, and so on [17–20].”

The subject of the reference [20] is not SLA process. This reference is focused on FDM process. “Yang, L.; Zheng, D.; Jin, G.; Yang, G. Fabrication and Formability of Continuous Carbon Fiber Reinforced Resin Matrix 572 Composites Using Additive Manufacturing. Crystals 2022, 12, 649.” Please explain the link between the Line 80 and this reference.

7.      In AM I recommend to use ISO / ASTM52900 – 15 Standard Terminology for Additive Manufacturing – General Principles – Terminology. Please mention the AM standard in the paper and defined the LFS process based on the standard.

8.      The References related to the orientation of the part in additive manufacturing should be improved. You can cite the following papers:

Chen, Y.; Lu, J. RP Part Surface quality versus build orientation: When the layers are getting thinner, Int. J. Adv. Manuf. Technol. 2013, 67, 377–385. DOI:10.1007/s00170-012-4491-7

Cazon, A.; Morer, P.; Matey, L. PolyJet technology for product prototyping: tensile strength and surface roughness properties, Proc. IMechE Part B: J Eng. Manuf. 2014, 228, 1664–1675. DOI:10.1177/0954405413518515

Udroiu, R. New Methodology for Evaluating Surface Quality of Experimental Aerodynamic Models Manufactured by Polymer Jetting Additive Manufacturing. Polymers 2022, 14, 371. https://doi.org/10.3390/polym14030371

9.      How about the 3D models of the specimens. What software was used and what is the accuracy of the stl file.

10.  Please explain how the fabricated resin samples are stick to the lifting platform. Is it necessary a support?

11.  The regression equation was obtained based on the experimental data. The least square method supposes an interpolation between the experimental data. The obtained regression equation is true only for this resin build, and in this printing condition. If the resin or the printing condition will be change the regression equation is different. Please explained the regression if the conditions will be change?

12.  The cuckoo search algorithm used in this paper should be briefly explained for the user understanding. What software was used for cuckoo search algorithm implementation?

13.  For each group of parameters of the printing orientation, 5 resin samples were prepared respectively and their mechanical properties were detected separately… A statistical analysis of the data should be done. How was verified the repeatability of the data?

14.  All the conclusions should be done based o the results. Also, future research directions may also be highlighted.

15.  Are the limitations of this study noted? The limitations of this study should be discussed.

Author Response

Response to reviewer 1

General Comment: In this paper, the authors investigate the mechanical properties of resin sample for different orientation, obtained by Low Force Stereolithography (LFS) on Form3 3D printer. Using the cuckoo search algorithm it was found the optimal part orientations with the best mechanical properties. From my point of view there are some aspects to improve.

Response:

Thank you very much for your kind review to our manuscript and the meaningful comment to our research. We have revised the manuscript carefully according to your and other reviewers’ comments. The responses to your comments are as follows.

  1. The title of the paper should be written according to the 3d printing process that was investigates in the paper. Instead of Stereolitography should be used Low Force Stereolithography to avoid confusion with traditional Stereolitography process.

Response:

Thank you very much for your kind suggestion. The title is modified as “Investigation and Optimization of the Printing Orientation on Mechanical Properties of Resin Sample in the Low Force Stereolithography Additive Manufacturing” in the revised manuscript.

  1. Abstract does not contain any quantitative data about the results. The abstract should be revised and improved.

Response:

Thank you very much for your significant suggestion. The abstract is modified to add the quantitative data to replace the qualitative results, and these modifications are highlighted in yellow in the revised manuscript.

  1. The structure of the paper is unclear for a good understanding of the reader. Thus the paper structure should be revised in order to clearly present the Material and Methods, and Results in separated sections. The Material and Method section should describe clearly the materials used and the proposed methods. The section Results should shown all the results obtained by applying the proposed method. Thus, Figure 5, Table 3, 4, 5 and 6 present results. Figure 6 and 7 describe the LFS process and they should be moved to the Material and Method section.

Response:

Thank you very much for your kind suggestion. The Tables 5 and 6 were moved to from the section ‘3. Modeling and Optimization’ to the section ‘4. Results and Discussions’, and the original Figure 6 ‘Schematic diagram of the fabrication process in the SLA’ was moved from the section ‘4. Results and Discussions’ to the section ‘2. Materials and Methods’, as shown in the revised manuscript. However, we suppose that the original Figure 5 ‘Cross–sectional experimental results of mechanical properties of the prepared resin samples.’ and the data in the Table 3 ‘The experimental data of mechanical properties for different printing orientations’ are the basic foundations for the theoretical modeling and parameter optimization in the section ‘3. Modeling and Optimization’, and it would be better exhibited before the section ‘3. Modeling and Optimization’, so neither of them were moved to the section ‘4. Results and Discussions’. Meanwhile, the Table 4 ‘Comparisons of prediction values with actual values.’ was part of theoretical modeling process, so it was not moved from section ‘3. Modeling and Optimization’ to section ‘4. Results and Discussions’.

  1. The Introduction section should be improved. It should define the purpose of the work and its significance. The current state of the research field should be carefully reviewed and key publications cited. Please highlight controversial and diverging hypotheses when necessary. Which are the conclusions from the literature survey?

Response:

Thank you very much for your kind suggestion. We have modified the introduction section according to your and other reviewers’ comments, which aims to make the manuscript more reasonable, and these corresponding modifications are highlighted in yellow in the revised manuscript.

  1. Line 34: “The metal additive manufacturing was..”. Please explain if there are any link of the metal AM to this research.

Response:

Thank you very much for your kind suggestion. We agree with you and other reviewers that the metal additive manufacturing method has little relationship with this research, so we replace these references by new references which have relationship with the resin 3D printing methods in the revised manuscript.

  1. Line 80: “For most of practical applications, the resin sample fabricated by SLA must have a certain mechanical property, which consists of the tensile strength, compressive strength, flexural modulus, and so on [17–20].”

The subject of the reference [20] is not SLA process. This reference is focused on FDM process. “Yang, L.; Zheng, D.; Jin, G.; Yang, G. Fabrication and Formability of Continuous Carbon Fiber Reinforced Resin Matrix 572 Composites Using Additive Manufacturing. Crystals 2022, 12, 649.” Please explain the link between the Line 80 and this reference.

Response:

Thank you very much for your kind suggestion. We have replaced the original reference [20] “Yang, L.; Zheng, D.; Jin, G.; Yang, G. Fabrication and Formability of Continuous Carbon Fiber Reinforced Resin Matrix 572 Composites Using Additive Manufacturing. Crystals 2022, 12, 649.” by “Kumar, M.; Ghosh, S.; Kumar, V.; Sharma, V.; Roy, P. Tribo-mechanical and biological characterization of PEG-DA/bioceramics composites fabricated using stereolithography. J. Manuf. Process. 2022, 77, 301–312.”

  1. In AM I recommend to use ISO / ASTM52900 – 15 Standard Terminology for Additive Manufacturing – General Principles – Terminology. Please mention the AM standard in the paper and defined the LFS process based on the standard.

Response:

Thank you very much for your kind suggestion. We have mentioned the recommend international standard in the revised manuscript for the description of the utilized Form3 low force stereolithography 3D printer in the section ‘2.1. Material Preparation’, and these modifications were highlighted in yellow.

  1. The References related to the orientation of the part in additive manufacturing should be improved. You can cite the following papers:

Chen, Y.; Lu, J. RP Part Surface quality versus build orientation: When the layers are getting thinner, Int. J. Adv. Manuf. Technol. 2013, 67, 377–385. DOI:10.1007/s00170-012-4491-7

Cazon, A.; Morer, P.; Matey, L. PolyJet technology for product prototyping: tensile strength and surface roughness properties, Proc. IMechE Part B: J Eng. Manuf. 2014, 228, 1664–1675. DOI:10.1177/0954405413518515

Udroiu, R. New Methodology for Evaluating Surface Quality of Experimental Aerodynamic Models Manufactured by Polymer Jetting Additive Manufacturing. Polymers 2022, 14, 371. https://doi.org/10.3390/polym14030371

Response:

Thank you very much for your kind suggestion. The recommended papers were added in the references in the section ‘1. Introduction’ in the revised manuscript. Meanwhile, the references are adjusted and modified according to your and other reviewers’ comments in the revised manuscript.

  1. How about the 3D models of the specimens. What software was used and what is the accuracy of the stl file.

Response:

Thank you very much for your kind comment. The three–dimensional model of sample was built in 3D modeling soft-ware Solidworks (Dassault Systèmes SOLIDWORKS Corp., Waltham, Massachusetts, USA) and it was saved as the .stl file. Afterwards, the .stl file was introduced into Preform software supported by Form3 3D printer, and its orientation could be adjusted relative to the reference coordinate system. There was no more special setting for the accuracy or the other parameters for the .stl file, which selected the normal setting in the Solidworks and Preform.

  1. Please explain how the fabricated resin samples are stick to the lifting platform. Is it necessary a support?

Response:

Thank you very much for your kind comment. As shown in the Figure 5 in the revised manuscript, there are special fixtures for each kind of detection of mechanical properties, which are provided to cooperate with the universal testing machine by Yangzhou Zhengyi Testing Machinery Co. Ltd, Yangzhou, China.

  1. The regression equation was obtained based on the experimental data. The least square method supposes an interpolation between the experimental data. The obtained regression equation is true only for this resin build, and in this printing condition. If the resin or the printing condition will be change the regression equation is different. Please explained the regression if the conditions will be change?

Response:

Thank you very much for your meaningful comment. We agree with you that the obtained regression equations in Equations (13), (14) and (15) are only for this resin build in this printing condition, and these regression equations will be different if the resin or printing conditions are changed. However, the results of experimental validation in the Table 4 prove effectiveness of the multiple regression model, which can provide meaningful guidance for the other printing conditions or the other additive manufacturing methods by adjusting the order and regression parameters in the multiple regression model.

  1. The cuckoo search algorithm used in this paper should be briefly explained for the user understanding. What software was used for cuckoo search algorithm implementation?

Response:

Thank you very much for your kind suggestion. Pseudo codes of standard Cuckoo search algorithm was added in the revised manuscript as the Table 5, which could explain the basic operating principle of Cuckoo search algorithm. The Cuckoo search algorithm was implemented in the Microsoft Visual C++.

  1. For each group of parameters of the printing orientation, 5 resin samples were prepared respectively and their mechanical properties were detected separately… A statistical analysis of the data should be done. How was verified the repeatability of the data?

Response:

Thank you very much for your kind suggestion. As mentioned in the section “2.1. Material Preparation”, in order to reduce the accidental error and improve the detection accuracy, 5 samples are fabricated in this study for each group of parameters of printing orientation, and the mechanical properties of each sample are detected separately. The final data of mechanical property is arithmetical average value of 5 experimental results. The testing results proved that for these 5 resin samples for each group parameters of the printing orientation, the obtained curves between load and deformation were basically consistent, and the derived differences among the 5 calculated values of each mechanical property were smaller than 4%.

  1. All the conclusions should be done based on the results. Also, future research directions may also be highlighted.

Response:

Thank you very much for your kind suggestion. We have modified the conclusions with your and other reviewers’ comments in the revised manuscript, and these modifications are highlighted in yellow.

  1. Are the limitations of this study noted? The limitations of this study should be discussed.

Response:

Thank you very much for your kind suggestion. As mentioned in the introduction section, there are many influencing factors in the additive manufacturing method besides the investigated parameters of printing orientation in this research, such as basic features of the photosensitive liquid resin, fabrication parameters, and post processing parameters. Moreover, there are other parameters for the additive manufacturing methods different from low force stereolithography. Therefore, more comprehensive considerations of the influencing parameters in various additive manufacturing method will be taken into account in the further research, which can be favorable to promote the applications of additive manufacturing method.

Reviewer 2 Report

The presented topic “Investigation and Optimization of the Printing Orientation on Mechanical Properties of Resin Sample in Stereo Lithography Apparatus Additive Manufacturing” concerns an important area of additive manufacturing. Article is written well and easy to understand. Article will attract more citations in the coming time. I have few comments as follows:

  • Need some of the quantitative findings in abstract
  • Too many keywords.
  • In introduction, it is necessary to give brief introduction about AM and its types. Following articles will be useful for the same: https://doi.org/10.1016/j.jmrt.2022.08.074 ;  https://doi.org/10.1007/s00170-015-7576-2
  • Briefly describe the summary of your work in last paragraph of introduction.
  • How the parameters were selected?
  • In results and discussion section, compare your findings with past studies and give reasonable agreement from literature
  • Conclusion: Firstly, add summary of your work in 2-3 sentences and then mention the outcomes with key results

Author Response

Response to reviewer 2

General Comment: The presented topic “Investigation and Optimization of the Printing Orientation on Mechanical Properties of Resin Sample in Stereo Lithography Apparatus Additive Manufacturing” concerns an important area of additive manufacturing. Article is written well and easy to understand. Article will attract more citations in the coming time. I have few comments as follows.

Response:

Thank you very much for your kind review to our manuscript and helpful assessment to our research. We have revised the manuscript carefully according to your and other reviewers’ comments. The responses to your comments are as follows.

  1. Need some of the quantitative findings in abstract.

Response:

Thank you very much for your significant suggestion. The abstract is modified to add the quantitative data to replace the qualitative results, and these modifications are highlighted in yellow in the revised manuscript.

  1. Too many keywords.

Response:

Thank you very much for your significant suggestion. The keywords are reduced to 6 in the revised manuscript and some secondary keywords are removed.

  1. In introduction, it is necessary to give brief introduction about AM and its types. Following articles will be useful for the same: https://doi.org/10.1016/j.jmrt.2022.08.074 ; https://doi.org/10.1007/s00170-015-7576-2

Response:

Thank you very much for your kind suggestion. The recommended articles are added in the references in the revised manuscript. Meanwhile, the references are adjusted and modified according to your and other reviewers’ comments in the revised manuscript.

  1. Briefly describe the summary of your work in last paragraph of introduction.

Response:

Thank you very much for your kind suggestion. A summary of our work is added in last paragraph of the section “1. Introduction” in the revised manuscript according to your comment.

  1. How the parameters were selected?

Response:

Thank you very much for your kind comment. As mentioned in the section “2.1. Material Preparation”, the investigated parameters of printing orientation relative to the x–axis, y–axis and z–axis was selected in the range of 0°–90° with the interval of 30°. Meanwhile, for the simulation process in the section “3.2. Parameter Optimization”, the range of α, β and γ were selected as [0°, 90°] with the interval of 1° by taking the fabrication accuracy into consideration. Moreover, for the other parameters in the 3D printing process except the investigated parameters of printing orientation, they were selected according to the corresponding national standards.

  1. In results and discussion section, compare your findings with past studies and give reasonable agreement from literature.

Response:

Thank you very much for your kind comment and significant suggestion. In the section “4. Results and Discussions”, the achievements obtained in this manuscript are compared with past achievements obtained in the literatures in the revised manuscript.

  1. Conclusion: Firstly, add summary of your work in 2-3 sentences and then mention the outcomes with key results.

Response:

Thank you very much for your significant suggestion. We have summarized the whole work in this research and added in the section “5. Conclusions” in the revised manuscript, and these modifications are highlighted in yellow.

Reviewer 3 Report

General Comment: The authors presented a study named “Investigation and Optimization of the Printing Orientation on Mechanical Properties of Resin Sample in Stereo Lithography  Apparatus Additive Manufacturing” which need to be improved with the following recommendations:

·         Abstract is not good, improve it with the numerical results and write it again succinctly. The main theme of this paper is not described in the abstract. Abstract section should be concisely reflected the content and summarize the problem, the method, the results, and the conclusions.

·         At the last paragraph of the introduction, please clearly show the general outline of the paper and show the importance of the study along with the main aim.

·         Please avoid multiple citations in the paper. Break these sentences into one sentence. Each one of the cited references  must be discussed individually and demonstrate their significance to your work.

·         Are all figures original? If not needed appropriate citations and permissions. Refine this for figures throughout the article. Are all formulas original? If not needed appropriate citations.

·         Describe the measurement procedure in more detail. At what point in time? How is the measuring setup set up? How many repetitions of measurements? What statistical methods are used to process experimental results? Describe the experimental stand in more detail. What method of experiment planning is used and why?

·         Indeed, there are an impressive amount of results. However, the conclusions section needs to improve with selected and highlighted main findings. In conclusion section, it is necessary to more clearly show the novelty of the article and the advantages of the proposed method. Add qualitative and quantitative results of your work. Please try to emphasize your novelty, put some quantifications, and comment on the limitations. This is a very common way to write conclusions for a learned academic journal. The conclusions should highlight the novelty and advance in understanding presented in the work.

·         Language used in the manuscript is generally satisfying. However, writers should pay more attention of singular / plural nouns. Also, they should control the spell check/ punctuation of words and sentences. In addition, spaces should be added between words and numbers.  Please check all manuscript for language and misspellings.  Please revisit all manuscript and correct such inconsistencies.

·         There is a reference problem. If your work is convenient for this journal’s context then there are many references from this journal.

·         The plotted values in all Figures or presented values in all tables need standard deviations (error bars).

·         The presentation is too wordy and lengthy and some figures are low quality in terms of scientific perspective. Some figures especially Figs. 1, 2, and 4 are trivial. Please revise them, you may merge them one figure for better scientific presentation. The authors did not present the paper with a clear logic flow and focus; instead they presented too much information that is not needed, or with limited value, or should be presented in a concise way.

·         The results section needs to be improved according to using proper citations and support the findings. Please improve all results with 5-6 lines with commenting on the figures. Present version in the paper lacks theoretical support. Please provide more detailed discussions to support author's discoveries on the research.

Author Response

Response to reviewer 3

General Comment: The authors presented a study named “Investigation and Optimization of the Printing Orientation on Mechanical Properties of Resin Sample in Stereo Lithography Apparatus Additive Manufacturing” which need to be improved with the following recommendations.

Response:

Thank you very much for your kind review to our manuscript and positive assessment to our research. We have revised the manuscript carefully according to your and other reviewers’ comments. The responses to your comments are as follows.

  1. Abstract is not good, improve it with the numerical results and write it again succinctly. The main theme of this paper is not described in the abstract. Abstract section should be concisely reflected the content and summarize the problem, the method, the results, and the conclusions.

Response:

Thank you very much for your significant suggestion. The abstract is modified to add the quantitative data to replace the qualitative results, and these modifications are highlighted in yellow in the revised manuscript.

  1. At the last paragraph of the introduction, please clearly show the general outline of the paper and show the importance of the study along with the main aim.

Response:

Thank you very much for your significant suggestion. We have summarized the whole work in this research and added in the section “5. Conclusions” in the revised manuscript, and these modifications are highlighted in yellow.

  1. Please avoid multiple citations in the paper. Break these sentences into one sentence. Each one of the cited references must be discussed individually and demonstrate their significance to your work.

Response:

Thank you very much for your kind suggestion. We have modified the cited references according to your and other reviewers’ comments, both in the text and in the reference list. However, limited by the length of the article and avoided the lengthy paper, not all the cited references are discussed individually. Among the 44 references in this manuscript, most of them are cited in the text and discussed individually, and some similar researches are conducted multiple citations, which is common in general research papers.

  1. Are all figures original? If not needed appropriate citations and permissions. Refine this for figures throughout the article. Are all formulas original? If not needed appropriate citations.

Response:

Thank you very much for your kind comment. All the figures are original. The pictures in the Figures 1, 3, and 5 are taken by ourselves, and the other figures in this manuscript are made by ourselves as well, so no citations or permissions are needed. The formulas of Equations (1) to (6) are fundamental formulas in the mechanics textbook, and these of Equations (7) to (12) are fundamental formulas in the mathematics textbook, so there will be no citations needed.

  1. Describe the measurement procedure in more detail. At what point in time? How is the measuring setup set up? How many repetitions of measurements? What statistical methods are used to process experimental results? Describe the experimental stand in more detail. What method of experiment planning is used and why?

Response:

Thank you very much for your kind suggestion. The measurement procedure are added in the revised manuscript. The added contents are as follows.

The measurement procedures of the tensile strength for each resin sample were as follows. Firstly, the clamp for tensile testing was fixed and positioning calibrated, which aimed to ensure the fitting accuracy between upper part and lower part. Secondly, the universal testing machine was turned on and the position limits were installed, which aimed to ensure the safety of the laboratory staff and that of the equipment. Thirdly, the resin sample for tensile testing was fixed in the clamp, and the upper and lower edges of the electronic extensometer were bound to the sample with rubber bands. Fourthly, the matched software in the computer was open and the plastic stretching procedure was selected. Meanwhile, the middle sectional size of the sample was input in the software, and the set system configuration was 1025E deformation sensor, automatic identification of fractures, preload force of 1 N, loading speed of 1 mm/min, and the full clearing of real-time data before next time testing. Fifthly, when the sample was fractured or the load reached the maximum value, the testing process stopped. Finally, the test data was exported and the residual sample was taken down.

The measurement processes of the compressive strength for each resin sample were as follows. Firstly, the clamp for compressive strength testing was fixed and positioning calibrated, which aimed to ensure the fitting accuracy between the upper part and lower part. Secondly, the universal testing machine was turned on and the position limits were fixed, which aimed to ensure safety of the laboratory staff and that of the equipment. Thirdly, the resin sample for compressive strength testing was laid flat on the center of the lower fixture, and position of the upper fixture was adjusted close to the resin sample without any direct contact. Fourthly, the matched software in the computer was open and compressive procedure was selected. Meanwhile, the size of the sample was input in the software, and the set system configuration was 1025E deformation sensor, automatic identification of fractures, preload force of 1 N, loading speed of 2.5 mm/min, and the full clearing of real-time data before next time testing. Fifthly, when the sample was crushed or the load reached the maximum value, the testing process stopped. Finally, the test data was exported and the residual sample was taken down.

The measurement procedures of the flexural modulus for each resin sample were as follows. Firstly, the clamp for flexural modulus testing was fixed and positioning calibrated, which aimed to ensure the fitting accuracy between upper part and lower part. Meanwhile, the span of the lower fixture was adjusted to 16 times of the width of the sample according to the national standard. Secondly, the universal testing machine was turned on and the position limits were fixed, which aimed to ensure safety of the laboratory staff and that of the equipment. Thirdly, the resin sample for flexural modulus testing was fixed on the lower fixture, and position of the upper fixture was adjusted close to the resin sample without any direct contact. Fourthly, the matched software in the computer was open and flexural procedure was selected. Meanwhile, the size of the sample and the span were input in the software, and the set system configuration was 1025E deformation sensor, automatic identification of fractures, preload force of 1 N, loading speed of 2 mm/min, and the full clearing of real-time data before next time testing. Fifthly, when the resin sample was broke or the load reached the maximum value, the testing process stopped. Finally, the test data was exported and the residual sample was taken down.

  1. Indeed, there are an impressive amount of results. However, the conclusions section needs to improve with selected and highlighted main findings. In conclusion section, it is necessary to more clearly show the novelty of the article and the advantages of the proposed method. Add qualitative and quantitative results of your work. Please try to emphasize your novelty, put some quantifications, and comment on the limitations. This is a very common way to write conclusions for a learned academic journal. The conclusions should highlight the novelty and advance in understanding presented in the work.

Response:

Thank you very much for your kind suggestion. We have modified the conclusions with your and other reviewers’ comments in the revised manuscript, and these modifications are highlighted in yellow. As mentioned in introduction section, there are many influencing factors in the additive manufacturing method besides the investigated parameters of printing orientation in this study, such as basic features of the photosensitive liquid resin, fabrication parameters, and post processing parameters. Moreover, there are other parameters for additive manufacturing methods different from low force stereolithography. Therefore, more comprehensive considerations of the influencing parameters in various additive manufacturing method will be taken into account in the further research, which can be favorable to promote the applications of additive manufacturing method.

  1. Language used in the manuscript is generally satisfying. However, writers should pay more attention of singular / plural nouns. Also, they should control the spell check/ punctuation of words and sentences. In addition, spaces should be added between words and numbers. Please check all manuscript for language and misspellings. Please revisit all manuscript and correct such inconsistencies.

Response:

Thank you very much for your kind suggestion. We have checked all manuscript for the language and misspellings, and these corrections are highlighted in yellow in the revised manuscript.

  1. There is a reference problem. If your work is convenient for this journal’s context then there are many references from this journal.

Response:

Thank you very much for your kind comment. We does not deliberately choose references from this journal or the other journals. All the references are selected for their contents to support the viewpoint in this study.

  1. The plotted values in all Figures or presented values in all tables need standard deviations (error bars).

Response:

Thank you very much for your kind suggestion. The experimental data in the Tables 3, 4, 6, 8 and 9 are added the standard deviations for the 5 experimental results of mechanical properties for each group of parameters of the printing orientation. For the plotted figures in the Figure 6, which are cross–sectional experimental results of mechanical properties of the prepared resin samples and they are used to explain how to derive the mechanical properties from the experimental data, so the error bars are not essential. Meanwhile, there are too many data points in the detection process, so the error bars are impossible to be added in the Figure 6.

  1. The presentation is too wordy and lengthy and some figures are low quality in terms of scientific perspective. Some figures especially Figs. 1, 2, and 4 are trivial. Please revise them, you may merge them one figure for better scientific presentation. The authors did not present the paper with a clear logic flow and focus; instead they presented too much information not needed, or with limited value, or should be presented in a concise way.

Response:

Thank you very much for your kind suggestion. Pictures of the used fabrication apparatus and detection system are required by the editors and other reviewers, which can improve reliability of this research and supply enough information for reexamination of the results in this study. Meanwhile, we have attempted to make the manuscript smoother and more readable by adjusting the whole text according to your and other reviewers’ comments.

  1. The results section needs to be improved according to using proper citations and support the findings. Please improve all results with 5-6 lines with commenting on the figures. Present version in the paper lacks theoretical support. Please provide more detailed discussions to support author's discoveries on the research.

Response:

Thank you very much for your kind suggestion. The section “4. Results and Discussions” is modified according to your and the other reviewers’ comments. We agree with you that there lacks mature theory to support the points in this research, so mechanism analysis based on the force analysis for variable conditions is conducted in this study and it can qualitatively reveal the reasons for differences in mechanical properties. We will conduct more research on the mechanism analysis of additive manufacturing in the future study, which aims to find the ideal theoretical support for the achievements in this research.

Round 2

Reviewer 1 Report

All the comments are addressed well and utilized to improve the manuscript. The manuscript is acceptable.

Reviewer 3 Report

The authors responded to all comments and significantly revised the manuscript. I think the manuscript can be accepted for publication.